# Community Organising Frameworks, Models, and Processes to Improve Health: A Systematic Scoping Review

**DOI:** 10.3390/ijerph20075341

**Published:** 2023-03-30

**Authors:** Shanti Kadariya, Lauren Ball, David Chua, Henriette Ryding, Julie Hobby, Julie Marsh, Karly Bartrim, Lana Mitchell, Joy Parkinson

**Affiliations:** 1School of Public Health, University of Queensland, Brisbane 4072, Australia; 2School of SHS–Nutrition and Dietetics, Griffith University, Gold Coast 4215, Australia; 3Australian eHealth Research Centre, Commonwealth Scientific and Industrial Research Organisation (CSIRO), Herston 4029, Australia

**Keywords:** community organising, frameworks, models, health

## Abstract

Community involvement engages, empowers, and mobilises people to achieve their shared goals by addressing structural inequalities in the social and built environment. Through this review, we summarised published information on models, frameworks, and/or processes of community organising used in the context of health initiatives or interventions and documented the outcomes following their use. A systematic scoping review was conducted in three databases with no restrictions on the date of publication, country, or written language. Out of 5044 studies, 38 met the inclusion criteria and were included in the review. The targeted health outcomes explored by the studies were diverse and included sub-domains such as the promotion of a healthy lifestyle, sexual and reproductive health, access to healthcare and equity, and substance abuse and chronic disease management. The outcomes of most initiatives or interventions were promising, with positive changes reported for the target populations. A wide variation was noted in the models, frameworks, or processes of community organising utilised in these studies. We concluded that variation implies that no single model, framework, or process seems to have predominance over others in implementing community organising as a vehicle of positive social change within the health domain. The review also highlighted the need for a more standardised approach to the implementation and evaluation of these initiatives. We recommend that it is essential to foster public and non-governmental sector partnerships to promote community-driven health promotion efforts for a more sustainable approach to these initiatives.

## 1. Introduction

Community involvement in research engages and mobilises people with an aim to achieve their shared goals by addressing structural inequalities in the social and built environment [1]. Communities that collectively take responsibility to influence and modify their social and built environment for the purposes of better health and wellbeing are increasingly recognised as effective vehicles for positive change [2]. Consequently, health promotion initiatives that genuinely engage communities are recognised as being more likely to succeed than those that do not engage communities [3,4,5].

Community organising primarily aims to bring people together to act on their common concerns; develop and expand their sense of community ownership; empower them; foster collaboration; and ultimately make the community powerful for the common good [6]. Community organising is an approach to community engagement that assumes communities can act on problems important to them collectively and make desired change. It also assumes the need for people to fully participate in managing organic change, thus supporting sustained positive change. Since these efforts to improve health are driven by community members rather than policymakers, health professionals or other stakeholders, the communities’ priorities take precedence [7,8].

Researchers from several disciplines, including health, have embraced community organising in their research and/or interventions, due to the increased likelihood of success when launched as community organising initiatives [4,5]. Many of those researchers have adopted one or more theoretical/research frameworks, models and community organising processes to structure, inform and/or guide their intervention(s). As a result, the current health literature on community organising comprises diverse initiatives that have adopted a wide range of frameworks, models, and/or processes of community organising. However, to our knowledge there is currently no published study that consolidates and summarises those initiatives and their frameworks, models, and processes in one place. Having this information consolidated will not only provide an opportunity to explore the variety of frameworks, models, and processes but also help identify how they were implemented, the gaps in their implementation, and the ways to strengthen their implementation in future health initiatives. 

Therefore, this scoping review will answer the following questions:(1)What models, frameworks, and processes have been used in community organising initiatives that advance health?(2)What target health behaviours or topics have community organising initiatives addressed?

## 2. Methods

This scoping review followed the Joanna Briggs Institute (JBI) scoping review guidelines [9] and has been reported in line with PRISMA-SCR (Preferred Reporting Items for Systematic Reviews and Meta-Analyses extension for Scoping Reviews) reporting guidelines [10].

## 3. Search Strategy

Relevant studies were searched in three databases, SCOPUS, Sociological Abstracts, and Google Scholar. Free texts, MeSH terms, and controlled terms were used to construct the search with help from a university health librarian. Search strategies were modified according to each database platform’s command language. The synonyms to indicate “community organising”, “community”, and “framework/theory/model” were used as search terms. For Google Scholar, we screened the first ten pages of results. The reference lists of all studies selected from those ten pages were also reviewed to identify additional relevant studies. 

Studies were eligible for the review if they met the following criteria:A clear reference to community organising as an essential aspect of the initiative.The community organising effort related to health, wellbeing, or a social determinant of health.A theory, model, or framework of community organising was specifically mentioned as being used to inform the work.

Peer-reviewed, grey literature, conference papers, and empirical or conceptual studies were considered for inclusion in the review. No limits were placed on the date of publication, study design, or the language of the studies.

## 4. Study Selection

All identified studies were exported from the databases to the reference management software Zotero (version 5.0.96.3) and transferred to the review data management software, Covidence. Upon importing the studies to Covidence, duplicates were screened and removed. The screening was performed in two stages. First, study titles and abstracts were screened in duplicate by all authors (SK, JP, DC, HR, JH, JM, KB, LM, LB) and clearly irrelevant studies were excluded. Second, full-text versions of each of the studies were assessed in duplicate by authors (SK, JP, DC, HR, JH, JM, KB, LM, LB) and those not meeting the criteria were excluded. The reasons for exclusion of full review studies were documented (see Figure 1). Any discrepancies and conflicts were resolved by discussion between two authors (LB, DC). The remaining authors were also involved to meet consensus if further disagreements arose. 

## 5. Data Extraction

Data extraction from the included studies was conducted by all authors using the Covidence data extraction function. A purpose made template was developed and included details of first author’s name, year of publication, source of the article, title of the article, study design, and methods, geographic focus, sector, focus (context/behaviour/movement), and the community organising models, frameworks, and/or processes reported in the paper. Before data extraction commenced, all members of the research team involved in this step piloted the template, and amendments to the table were made according to their feedback. 

For each paper, data were extracted independently by two researchers and the results were compared via Covidence. In case of disagreements, the consensus was met through discussion or consultation with a third reviewer. 

## 6. Data Analysis 

The data extracted from the included papers were narratively summarised and accompanied by descriptive tables. 

## 7. Quality Appraisal

The mixed methods appraisal tool (MMAT), version 18, was used to critically appraise the articles [1]. This tool was specifically designed for systematic mixed studies reviews. The tool helps to examine the methodological quality of qualitative research, quantitative descriptive research, and mixed methods using relevant quality questions out of five key criteria. For each selected full-text papers, researchers (SK, DC, JM, JH, HR, LM, KB) reviewed quality and suitability, including appropriateness of study aim, adequacy of the methodological approach, representativeness of target population, data analysis, presentation of findings, and authors’ discussion and conclusion. Discrepancies were resolved through discussion.

## 8. Results

A total of 5044 studies were primarily identified in the search and 348 studies were automatically removed by Covidence as they were duplicates. The remaining 4696 studies were then screened by their titles and abstracts. After excluding 4546 non-relevant studies (unrelated to the review’s outcome and explanatory variables), 150 full text studies were reviewed. Of these, 38 met the inclusion criteria and were included in the review. 

The main reasons for exclusion were: the study scope not falling under health domain (n = 72), not related to community organising (n = 27), no framework, model, theory discussed (n = 8), wrong study design (n = 3), wrong intervention (n = 1), and wrong outcomes (n = 1). Of the 38 studies selected for final analysis, 33 were conducted in the United States [2,3,4,5,6,7,8,9,10,11,12,13,14,15,16,17,18,19,20,21,22,23,24,25,26,27,28,29,30,31,32,33,34], two in the United Kingdom [35,36] and one each in Canada [37], Japan [38] and South Africa [39]. The target populations of 11 of these studies were children and youth [3,6,20,21,22,23,25,29,32,34,40] followed by six studies, each with general [16,26,27,31,35,39] and marginalised population groups [2,5,13,14,15,19]. Other target population groups were older adults [9,10,37,38,39], members of community organising teams themselves [11,36], parents or families [4,7], at-risk or vulnerable populations [28,33], gender minorities [17], and health care professionals/institutions [24]. The studies included in the review were conducted between 1978 and 2021, more than half of the studies 63% (n = 17) were conducted in the last ten years [2,3,4,5,7,10,13,14,18,19,20,21,24,25,27,28,29,30,32,33,34,36,38,40], 21% (n = 8) were conducted in the last 20 years [6,9,12,17,22,26,35,37], and 15% (n = 6) were conducted prior to that [11,15,16,23,31,39] (See Table 1).

The most common study design was a case report or case study (n = 9) [3,13,14,16,25,28,30,33,34] and qualitative design (n = 9) [5,11,15,19,21,24,26,36,37], followed by seven studies conducted as published project implementation and evaluation [4,9,10,17,18,29,39]. Four studies were prospective or longitudinal in design [6,23,27,32], including cohort studies, and another four had a quasi-experimental design [7,35,38,40]. Three studies were community randomised trials [20,22,31]. The remaining studies had adopted participatory research [12] and a community listening exercise [2]. 

In most studies, community organising was undertaken by either a charity, a not-for-profit organisation, or a non-government or civil society organisation [3,5,7,13,17,18,19,20,26,28,29,34,36,37]. University researchers were the second most common group of community organisers [9,10,15,16,25,27,31,32,38], followed by government agencies [2,4,21,24,40]. A diverse group of organisers including community groups or leaders, health professionals, service managers, or a combination of several communities and/or institutional actors were involved in some studies [6,14,23,33,35]. In the remaining studies, it was not clearly stated who the community organisers were [11,12,22,30,39] (See Table 1).

The length of time required to undertake the preparatory work at the start of the action phase was reported by 19 studies only. Most studies took 24 months or more to start the action phase [3,15,29,30,31,32,34,38], while five took more than 12 months but less than 24 months to do so [16,27,35,36,37]. Three studies took 6–12 months [13,18,24], while two took between 1 and 6 months [21,33]. Only one study was found to have initiated the action phase within less than a month following the listening and organising process to initiate its community action [19].

### 8.1. Frameworks, Models, and Processes Adopted 

The review identified 22 different frameworks, models, or processes adopted by the studies (see Table 2). A framework usually represents a structure, overview, system, or plan composed of descriptive categories and does not provide an explanation, rather it groups the empirical phenomena into a set of categories. A model, on the other hand, is descriptive and typically involves a deliberate simplification of a phenomenon. Finally, a process is the analytical representation of the program activities. There were 10 studies that explicitly reported the use of a community organising framework [5,6,9,24,27,28,31,32,34,35]. All studies reported their use of community organising steps, and these varied between four and 10 steps. For instance: Cheadle et al. (2009) utilised community organising in the promotion of physical activity among older adults from southeast Seattle in the US. This study identified and involved champions in partner organisations for support and resources [9]. Rask et al. (2015) utilised the community organising technique to assess provider engagement and its impact in addressing the root cause of preventable readmissions by identifying participant-defined barriers [24]. Bezboruah (2013) examined the community organising technique to promote accessible and affordable health care to a marginalised neighbourhood in a large and diverse community [5].

Similarly, Mckenzie et al. (2004) used community organising to create a community-wide cancer education/screening program [35]. The study selected local leaders based on their previous professions and contribution to the community in order to assess community needs, followed by the implementation and evaluation of the program [35]. Zanoni et al. (2011) employed community organising to address the epidemic of asthma and obesity among Latina/o children. They motivated their parents to create knowledge, take action, reflect on outcomes and have their voice heard in dominant-culture schools [34]. Santilli et al. (2016) were guided by community organising principles to mobilise community members and partners to develop and build community support for neighbourhood-driven intervention in chronic disease prevention [27]. A community organising approach to counter alcohol abuse through a community randomised trial was outlined by Wagenaar and colleagues [31]. Bosma et al. in 2005 also adopted a community organising model to prevent substance use and violence among young adolescents in school settings [6]. Wagoner et al. (2010) employed a practice-based community organising conceptual model using a grounded-theory approach [32]. 

Each of the above-mentioned studies conducted a needs assessment as a part of their community organising process. One-to-one networking with the community members and local organisations was the most followed approach for needs assessment. Some of these studies also hired community organisers to assist [9,27]. However, only two studies evaluated their achievements or goals at the end of the program [31,35], and only one had a sustainability strategy in place [24]. Community members were empowered to address the local issues of alcohol, tobacco and violence [6]. The logic behind emphasis on the community’s active role in the process was explained by Saxon et. al. [28], who stressed that participating in community organising tends to give community members more ownership over local issues and maintains power balance between the community members and community organisers. 

#### 8.1.1. Collective Impact Framework

Collective impact of some kind was employed as the model or framework of community organising, either on its own or as a part of multiple framework structure in at least four studies [18,22,23,26] (See Table 2). The Collective Impact Framework is a collaborative approach centred on the tenet that to create long-term change for complex social and health issues, organisations must coordinate their efforts around a common goal [41]. There are five core components of the Collective Impact Framework including a common agenda, shared measurement systems, mutually reinforcing activities, continuous communication, and a backbone support organization [42]. 

In a study conducted by Poole and Colleagues in 1997, collective impact was set within the broader framework of the utility of action structures and was aimed to be a driver towards the attainment of national goals through a bottom-up approach in Oklahoma, US [23]. Salem et al. (2005) used the framework to increase the capacity of communities in Chicago, US to participate in public health decision making, promote new partnerships, make decision makers accessible to the communities, and to find a suitable role for the local public health department to support community-based health activities [26]. Hilgendorf et al. (2016) evaluated the lessons learned from a pilot obesity prevention approach by the Wisconsin Obesity Prevention Initiative for community action, which incorporated coalition as well as community organising efforts in two counties with the long-term goal of empowering community leaders to drive ongoing action [18]. Perry et al. (2000) represented the second phase of an early adolescent intervention against students’ alcohol use. This study promoted community action to increase community efficacy, integration and resilience to bring about positive changes in community norms [22]. 

These studies utilised a community-driven approach and actively involved, engaged or empowered community members, organisations and stakeholders in problem identification, development of solutions, planning and implementation of intervention(s). Community mobilisation activities were implemented and comprised the use of media campaigns and public events [22]. In addition, seeking support from community leaders in raising awareness of the problem and encouraging community members to take action were their key features. Building coalitions with the community members and sharing responsibilities with them to solve local problems were important power approaches used by Poole [23] and Salem et al. [26]. Furthermore, Salem argued that local needs are community owned and meeting them is not a sole responsibility of any organization. 

#### 8.1.2. Alinsky-Style Organising IAF Framework

Two studies adopted Alinksy-style organising as their framework of choice [17,29]. Subica et al. (2016) adopted the Alinsky-style community organising (also known as Industrial Areas Foundation (IAF style organising) framework to summarise the community organising initiatives of several grant recipients of projects targeting childhood obesity-causing structural inequities within 21 culturally/ethnically diverse communities through the creation of 72 environmental and policy solutions [29]. Prior to this in 2003, Hays et al. had employed the same model to build a strong, supportive young gay and bisexual men’s community through different outreach events and team performances where they could protect and support each other, promote having safer sex and help in HIV prevention efforts as a bigger goal through their Mpowerment project [17]. 

The Alinsky model begins with “community organising” and is based on a concept of separate public and private spheres [43]. In the Alinsky model, power and politics both occur in the public sphere, and Alinsky argued that poor communities could gain power through public sphere action, which involves taking public action such as protests and occupation to shift control from entities such as local government to the community [43]. The studies adopting IAF as their guiding framework [17,29] used a combination of community organising and other methods such as focus groups, interviews, and community meetings to engage community members in the research process. Feedback was gathered from community members on the acceptability and feasibility of their interventions. Outcomes focused on changes in both health behaviours and policy change. Aligned with the Alinsky approach, Hays et.al. [17] reported on shared power through the involvement of a core group who represent the various segments of the target community. This also signifies one way of maintaining the power balance between community organisers and community “organisee”. 

#### 8.1.3. Community-Based Participatory Research Framework

Three studies adopted community-based participatory research framework (CBPR) in their community organising effort [3,19,25]. The proponents of community-based participatory research (CBPR) claim that it benefits community participants, health care practitioners, and researchers alike [44]. They argue that CBPR creates bridges between scientists and communities through the use of shared knowledge and valuable experience [45,46]. Therefore, CBPR is a collaborative approach that emphasises the active involvement of community members in all facets of the research process [19]. All studies embracing CBPR [3,19,25] prioritised community ownership, active involvement of community members, and addressing potential biases in different ways. Following the CBPR framework, community ownership and active involvement were achieved through implementing community dialogue, community problem-solving, participatory action research and social justice approaches. Kang [19] and Bauermeister [3] highlighted the shared power achieved through collaboration with community and researchers while considering the geographical diversity. Similarly, Ross et al. [25] described the power of youth’s research and participation/action on health behaviour change to inspire key decision makers for developing new policies and ordinances. 

#### 8.1.4. Socio-Ecological Framework

Two studies that adopted the socio-ecological framework also used a community-based approach [10,33]. One of these two studies aimed to understand the structural issues that affected the creation of a community action and advocacy board (CAAB) and to identify strategies for overcoming those issues [33], while the other aimed to promote physical activity in older adults [10]. Community engagement and empowerment were central to both studies. Power was emphasised in both studies. For example: Weeks [33] recognised that members of the community were grassroots “experts” and “leaders” in key areas of HIV/STIs prevention, women’s health and empowerment, and community health needs. Cheadle [10] discussed the involvement of community in coalitions, thereby promoting the bottom-up approach.

A range of additional frameworks were applied as a part of community organising in the remaining studies. Coalition between different community partners, formative research, action planning and mobilizing the community members to solve the local needs by utilizing community resources, implementation and dissemination are major focus areas of these additional models or frameworks, including Rothman’s development model [16], innovative program planning frameworks [8], implementation science frameworks [4], community partnership frameworks [7], community need assessment methodology model [2], and communities mobilizing for change model [20]. Green care theory was one of the standalone theories that did not fit under one broad theme and it was used to describe connectedness to the natural environment by a collaborative approach [36]. 

The Mckenzie and Smeltzer community organisation model [37], Cottrell’s Community Competency framework [11], and the eight steps process [39] were three models that were bound by the commonality of using the eight step process. In this process, the initial steps were focused on community contacts and one-to-one relationships with individuals and groups to identify barriers and assets in the community. The intermediate steps facilitated the open meeting to identify the community needs and their solutions. The final steps, on the other hand, were focused on maintaining and sustaining the program. 

Citizen health care model [12], Freire’s theory of adult education [15], and Community capacity framework [21] were other models with the common aim of emphasising the knowledge, wisdom, and energy of an individual to promote their full participation in healthcare as a coproducer rather than just a patient or consumer. They believed that professionals can serve as catalysts in fostering such citizen initiatives with program ideas and directions through the creation of community partnerships and coalitions. Therefore, they emphasised power between professionals and consumers by enhancing individual capacity to command/demand more control over their health.

Similarly, Tawtaw used a conceptual framework that combines organizing theory and horizontal participatory approaches for the community health improvement plan life cycle [30]. Fawcett et al. used three playbooks for implementing an organizational change model, where initial briefing were carried out with partners who then reviewed the recommendations and developed an action plan for implementation [14]. Haseda and colleagues [38] visualised and figured out community health needs by using a JAGES Health Equity Assessment and Response Tool and developed community diagnosis forms [38]. Furthermore, Haseda [38] and Fawcett [14] both discussed empowering the local health sector and engaging them through coalition.

### 8.2. Targeted Health Behaviour or Topics Used by Community Organising Initiatives

Non-specific health promotion, health education, or lifestyle modification were the primary objectives of community organising in most studies [2,10,11,16,19,23,26,27,36,39]. This was followed by substance use (alcohol, tobacco, illicit drugs) and violence and associated behaviours [6,20,22,25,31,32]. Chronic disease management was the focus of four studies [13,21,29,35] including a cancer education/screening program in the UK, childhood obesity among low SES communities in the UK, asthma in Detroit, Michigan and obesity in the US. Health care access was the focus of another four studies [14,15,24,38], with one study focusing specifically on culturally appropriate healthcare access [14]. The other areas of specific focus were sexual/reproductive health [3,7,17,33], healthy eating [15,37], cancer prevention [35], gender discrimination [28], local environment factors [21], physical activity [9], social activities of older adults [38], assessment of community partnership [30], and coordination of health services [24]. However, it should be noted that many of these studies had overlapping focus areas especially relating to health promotion/healthy eating/healthy lifestyles. The outcomes of most initiatives were promising, with positive changes reported (at least in the short term) in health outcomes for the target populations in most studies (32/38) (see Table 1). 

#### Quality Appraisal

Out of 38 included articles, 32 were assessed using MMAT criteria (see Table 3). Six articles were excluded from MMAT assessment following the first two screening questions. The methodological quality of the studies was mixed. The quality of the qualitative articles was high, with 13 out of 16 studies meeting the five MMAT criteria. The mixed methods study was also of high quality and met all five criteria. Only two quantitative articles (2/14) met all possible appraisal criteria. Low non-response bias risk and blinded outcome assessors to the intervention were the most frequent unmet criteria for quantitative studies.

## 9. Discussion

This review synthesised the literature on community organising initiatives that pursued advancements in health. The review aimed to identify the targeted health behaviour or topics that community organising initiatives have addressed as well as models, frameworks, and processes that have been used by those initiatives. Overall, the review found that community organising has been regularly utilised over several decades as a guiding mechanism for community-based health initiatives. Positive changes were reported in health outcomes for the target populations in most of these initiatives.

Despite the use of community organizing frameworks over several decades, there is still no single gold standard framework adopted. A wide variety of models, frameworks, or processes of community organising were applied in the included studies. The variation implies that no one specific model, framework or process seems to have predominance over others in implementing community organising as a vehicle of positive social change within the health domain. Some frameworks that were common between studies that reported positive outcomes were the community organising model [6,10,32], socio ecological model [9], Rothman’s locality development model [16], and community-based participatory research model [19,25]. Despite such a wide variation, some themes were prevalent across the reviewed studies, including (1) the creation of partnerships and coalitions, (2) community integration and resilience, (3) joint problem-solving, (4) bottom-up approach, (5) community ownership, (6) community empowerment and (7) capacity building. Therefore, regardless of which framework is used, health interventions or initiatives are likely to deliver positive outcomes if they are delivered in a coordinated manner by incorporating these core components. As a result, future research should focus on supporting these key components to be a more common part of community activities.

Most studies (33/38) included in the review were conducted in the United States. This strong adoption of community organising could be because community organising as a vehicle for change began earlier in the United States than in other countries, potentially in Philadelphia with the wages strike in 1786 [47]. Additionally, the dominant political ideology in the United States (e.g., desire for small governments and lower taxes etc.) along with a largely inequitable, predominantly user-funded healthcare system could be seen as further drivers for more community organising activity since it allows people to organise, unionise, and consolidate their power [48,49]. Countries such as Australia and the United Kingdom, which can be considered comparable to the United States in many aspects such as language, culture, democratic election of government, etc., have also had a long history of labour unions, but these countries are more distinct from the United States due to their publicly funded healthcare systems (Medicare and National Health Services respectively). The health systems in these countries are known to be more equitable [50,51], thereby limiting the need for citizens and communities in these countries to organise for access to healthcare. Despite these factors, there are gaps which could be filled by more community-focused and community driven health initiatives or interventions. However, this review suggested that community organising as a vehicle of health initiatives or interventions has yet to pick up traction in countries outside of the United States. Therefore, it highlights the opportunity for the concept to be expanded in public health initiatives outside of the United States, learning from the experiences of the studies implemented there.

Most studies targeted a broad, general population, while some focused on specific population groups. Despite the heterogeneity in the selection of target population groups, there was consistency among most studies in terms of positive change reported in their targeted health outcomes. Such consistency in positive outcomes despite the variation among target population groups reinforces the argument that community organising has the potential to be an important vehicle for positive change [2,3,12,17,19]. This notion is important, particularly for the marginalised and disadvantaged communities who are more likely to be overlooked by existing mainstream health initiatives or interventions that are perhaps designed using a one-size-fits all approach [52]. Community organising provides an opportunity to listen to community voices and concerns, engage them deeply, work together with them to address those concerns, and create solutions to the community problems together with them, instead of adopting top-down approaches [52]. Authors have also referred to this bottom-up approach as a means of power-sharing with the communities to help them solve their own problems [53]. Power sharing has been recommended as an essential driving force and strategy behind other grassroots community initiatives [54,55]. However, when examining power issues, there are likely to be evaluation challenges. For example, how to measure shared power, Kang [27] and Bauermeister [3], or Ross et al. [25], power of youth’s research and participation/action on health behaviour. While there a range of frameworks identified in this review, they are applied to different contexts. Future research could examine the suitability of different frameworks for different community contexts, taking into consideration their unique issues and starting points. 

The reviewed studies did not document long-term outcomes or health impacts. While most studies reviewed reported positive change in the health outcomes, we noted that these measures were typically collected over relatively short and focused project durations. Many studies did not discuss the long-term sustainability of positive impacts, particularly after the funding had been exhausted. Only 18/38 articles in the review mentioned prolonged action or sustainability of community organising efforts beyond their research period. Some of the major strategies employed to sustain the community organising actions beyond the funded period in these studies were: empowering the community leaders and educating community members to engage and maintain the community action; continuing ongoing meetings with stakeholders; and ensuring trust between them and the community organisers. Some articles also discussed influencing public policy change/government support as a strategy to sustain community organising initiatives beyond the life duration of a particular focused project [5,22,31]. Utilising “partnership brokers” such as local governments and non-governmental organisations has also been suggested in initiatives beyond the health sector [56] to ensure the sustainability of community-institution partnership through the establishment of a systems-based approach. Studies included in this review in terms of outcome selection can be categorised into four groups: (1). to identify issues (1/38); (2). description of program implementations (6/38); (3). evaluation of program implementations with lessons learnt and influencing factors (19/38); and (4). effect directly on health outcomes (12/38). However, among the 12 studies directly reporting on health outcomes, five discussed quantitative/statistical conclusions. These four categories show the outcomes with increasing correlation to the ultimate objective: to improve health. Quality of implementation and effect on community capacity is an intermediate outcome to the final health outcome. The inclusion of more direct health measurements would improve the ability to evaluate the impact of these initiatives. Future studies should aim to measure long-term impact from their initiative, not just the measurement of outcomes during the funded period. 

## 10. Strengths and Limitations of the Review

This review brought together evidence on the use of community organising in the health domain and the adoption of several frameworks of community organising. However, the review did not systematically assess whether the studies adhered to the framework guidelines in a step-by-step manner. Therefore, the review should not be viewed as an assessment of their level of adherence to these frameworks. This could also be considered this review’s strength as it identified that there is not any standardisation/guideline for reporting such adherence. Another limitation of the study lies in the heterogeneity of topics and community groups, meaning that a rigorous meta-analysis was not possible. Nonetheless, the heterogeneity (of target population groups and frameworks) can also be considered a strength as it is suggestive that the approach can be used in many contexts and is therefore worthy of further consideration. Most studies included in the review also showed that the initiatives successfully improved the targeted health outcomes in the short term, which indicates the positive role of community organising in solving community problems with their active involvement. It needs to be acknowledged that it might also be reflective of selective publishing, where relatively less successful initiatives are not published and of a lack of follow-up studies to check whether these successes sustain over time in the absence of an active implementation team of community organisers/organisations. The assessment of publication bias and sustainability assessment was beyond the scope of this review.

## 11. Implications for Research

No guidelines exist to inform the development or reporting of tools to implement and evaluate community initiatives or interventions in a consistent way to enable comparison and conclusions to be drawn. Therefore, future studies could emphasise developing such implementation and evaluation guidelines to support the implementation and assessment of implementation fidelity and allow for comparability across initiatives. Furthermore, assessing the sustainability of community organising initiatives beyond the short-term project duration will also be helpful, since a lack of funding and active engagement from community organisers might mean that programs are discontinued, and any community benefits gained could cease or even regress. 

## 12. Conclusions

This review showed that community organising is a promising approach to community-based health initiatives. Health initiatives with successful outcomes in recent decades include the widespread shift to a bottom-up approach towards including community members in organising efforts to address their identified needs through active participation in their community. There is opportunity for a more standardised approach of implementation and evaluation of these initiatives, including objective measures of success and long-term sustainability. Future research should explore whether long-term sustainability can be achieved by encouraging a more proactive public sector role or by fostering a public- and non-governmental sector partnership to promote community-driven health promotion efforts. Regardless of the approach, ensuring community trust and empowering local leaders should remain the cornerstone of all these initiatives. 

## Figures and Tables

**Figure 1 ijerph-20-05341-f001:**
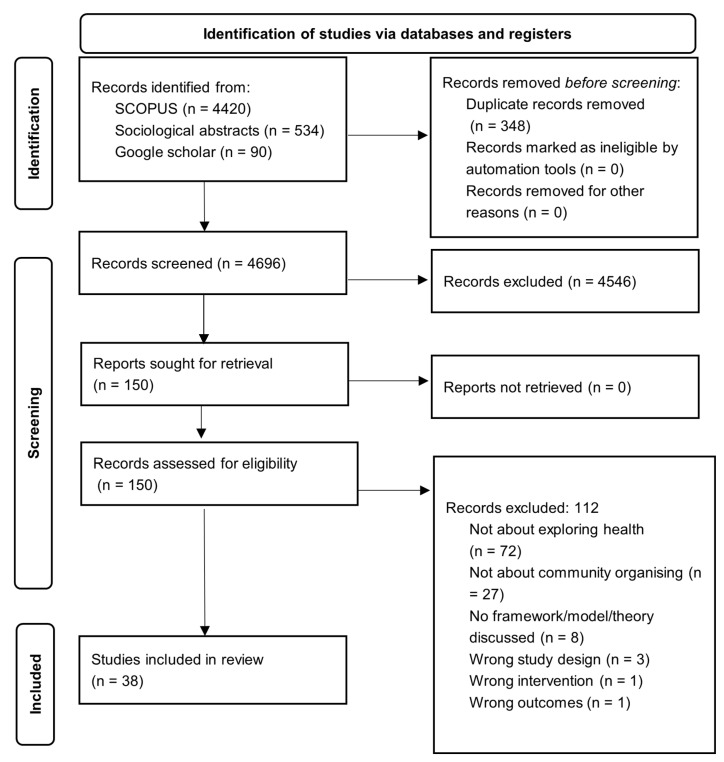
PRISMA flowchart detailing the article selection process for the scoping review.

**Table 1 ijerph-20-05341-t001:** Sample characteristics, methods and outcomes of the reviewed studies on community organizing and health (n = 38).

Study No.	First Authors Years	Country	Study Design	Target Population	General Focus/Foci of the Community Organising	Community Organising Model/Framework	Stated Aim(s)/Objective(s)	Outcomes/Impacts	Intervention Success *
(1)	Agrusti 2020 [2]	United States	Survey and community listening sessions	Marginalised neighbourhoods/communities/groups	Health promotion (non-specific)	Community needs assessment methodology model	To explore the social determinants and community needs for behavioural health services in socio-economically disadvantaged neighbourhoods	Community needs assessment showed residents’ inability to access essential mental health services. Community listening sessions revealed how behavioural death disorders had made life difficult for lower-income residents.	Yes
(2)	Bauermeister 2017 [3]	United States	Case report	Young adults	Sexual/reproductive health and prevention	Community-based participatory research	To develop a structural initiative for reducing HIV/STIs among YGBMTW in Southeast Michigan; and to identify and implement innovative community strategies to reduce STIs experienced by young men who have sex with men in the region.	The Health Access Initiative (HAI) and the Advocacy Collective (AC) were implemented after community listening. The initiatives developed multi-media resources and workshops for medical and social service providers, health educators, and policy-makers interested in providing youth-friendly services to sexual and gender minorities.	Yes
(3)	Berman 2018 [4]	United States	Project evaluation	Families	Obesity; Healthy lifestyles (non-specific lifestyle prevention/modification)	Implementation Science Framework (Proctor and colleagues)	To evaluate the Healthy Lifestyles Initiative, which aimed to strengthen community capacity for policy, systems, and environmental approaches to healthy eating and active living among children and families	Eighty initiative partners completed a brief online survey on implementation strategies engaged in, materials used, and policy, systems, and environmental activities implemented. The use of materials was positively associated with the implementation of targeted activities.	Yes
(4)	Bezboruah 2013 [5]	United States	Qualitative research	Marginalised neighbourhoods/communities/groups	Healthcare access and equity	Community organising conceptual framework	To highlight the lessons on community organising for health care in a diverse county	A lack of equal participation and consensus was faced during intervention implementation. A lack of statutory pressure with adequate funding to implement programs was also felt. Despite challenges, the goals of building trust and generating commitment from community members was achieved through formal and informal mechanisms.	NA +
(5)	Black 2020 [36]	UK	Qualitative research	Volunteers and staff at the garden	Healthy lifestyles, wellbeing, and local environment factors	Green care theory by Cutcliffe and Travale (2016)	To stimulate discourse around the role of community gardens in promoting social and environmental change for well-being	The community participants contributed to the project after knowing it was a part of the development of a play. The study also found that connectedness with the non-human elements of the environment contributed positively to wellbeing of the participants.	Yes
(6)	Bosma 2005 [6]	United States	Cohort study	Young adults	Substance use	Community organising model	To describe the community organising methods and process results of intervention to prevent substance use and violence among the youth	The organising component of the project was successful in engaging adults and youth in prevention activities in schools and communities. Positive behavioural results were achieved for boys who had lower rates of increase in cigarette and alcohol use and violent behaviours.	Yes
(7)	Brookes 2010 [7]	United States	Quasi-experimental	Parents	Sexual/reproductive health and prevention	Community organisation and community partnerships	To increase openness, acceptance, and engagement on issues of sexual health with parents of children ages 8–18 years	“Sex ed for parents” program was found to be useful to parents as they felt more comfortable talking about sexual health in their community. Parents found new ways to help their child with issues of intimacy, relationships and sex and provided them with information they wanted to share with other parents in their community. Overall, the Real Life. Real Talk. initiative managed to build grassroots partnerships; to reduce stigma and shame associated with talking about healthy sexuality; and to increase openness, acceptance, and engagement on issues of sexual health.	Yes
(8)	Bryant 2010 [8]	United States	Quasi-experimental	Children and adolescents	Obesity; Physical activity; Other: Obesity prevention	Innovative program planning framework, community-based prevention marketing	To show how an innovative program planning framework, community-based prevention marketing (CBPM), was used to design a physical activity promotion intervention, VERBTM Summer Score-card to improve physical activity opportunities for tweens (9–13 years old).	The VERB TM Summer Scorecard program was developed and adopted in or tailored for at least 15 additional communities. Female participants were found more likely to be physically active 4 or more days per week (29% ((OR = 1.29, *p* = 0.040)} than non-exposed and non-participating girls and 42% (OR = 1.42, *p* = 0.0041) more likely to be physically active 6 to 7 days per week in 2004. Three years later it increased to 80% (OR = 1.80, *p* < 0.0001), 53% (OR = 1.53, *p* < 0.0001), and 46% (OR = 1.46, *p* < 0.0001) more likely to be physically active 2 days or more, 4 days or more, and 6 to 7 days, respectively compared to non-participatory girls.	NA +
(9)	Cheadle 2010 [10]	United States	Implementation and evaluative of partnership and networking	Older people	Physical activity (PA)	Social ecologic model	To describe a community organising approach to promoting physical activity among underserved older adults in southeast Seattle	New senior exercise programs were created including walking groups and Enhance Fitness classes. Ten new walking groups were started in a variety of community settings. The project organiser worked with staff at Parks and Recreation to identify sites, do promotion and outreach, and in some cases lead the walking groups. Low-income seniors started being part of the new PA program. A new community health coalition was formed with senior PA as a major area of focus.	Yes
(10)	Cheadle 2009 [9]	United States	Project evaluation	Older people	Physical activity	Community organising model	To report final SESPAN evaluation findings and lessons learned during implementation	‘SESPAN (The Southeast Seattle Senior Physical Activity Network) networking among organizations led to the creation of a number of senior physical activity programs that continue to serve previously underserved, low-income, multicultural communities in Southeast Seattle. In addition, the health coalition established through the SESPAN project, HARVC, has the potential to continue to generate new, sustainable programs and environmental changes. Most of these physical activity programs have secure funding sources and organizational support that give them a good chance of being sustained.’	Yes
(11)	Denham 1998 [11]	United States	Qualitative research	Qualitative study of community organisers	Health promotion	Cottrell’s Community Competency framework with eight areas of focus	(a) To examine how community organising was being used for health promotion efforts in rural areas (b) To explore mechanisms of improving community competence through community organising	Through 11 in-depth interviews with community organisers in rural areas of North Carolina, this study summarised several ways to increase dimensions of community competence. Factors such as promoting face-to-face interactions between community members, maintaining community control, making space for skills training makes way for success. It was also found that community organising promote communication and helps maintain community’s control over decision-making.	NA +
(12)	Doherty 2006 [12]	United States	Participatory research	Adult patients with diabetes, Children and adolescents (with diabetes), First nations people of the USA	Culturally appropriate healthcare access, Other-diabetes management, better well-being	Citizens Health Care	To describe the origins of the citizen health care model, its core tenets and practices, and examples of how this model has been applied in community settings.	Health behaviour changes were observed, majority of “support partnership” were successful in changing health behaviours positively. The Health Maintenance Organisation is considering expanding the partnership program to more clinical sites. Motivation to adopt healthier lifestyle was observed amongst patients who participated. Improvements in cultural awareness amongst clinical researchers involved. Families and health care providers had fellowship, education and support every fortnight. These meetings all started with members checking and recording each other’s blood sugars, weights and conducting foot checks. Culturally appropriate meals and nutrition education was provisioned for at meetings. Other activities also include exercise, education, and support sharing.	Yes
(13)	Douglas 2016 [13]	United States	Case study	Case study 1—low-income communities Case study 2—Asian immigrant and refugee communities	New & expectant mother support, breastfeeding, infant health, childhood obesity and wider health issues	Communities Creating Healthy Environments (CCHE) Change Model and Evaluation Frame	To address proximal and distal determinants of childhood obesity in low-income communities through community organising focused on organizational capacity building and community empowerment	Case 1—The report was viewed more than 1500 times. They were able to raise awareness by getting the Jackson Health System to promote breastfeeding. Case 2—community base increased from 200 to 377 and number of leaders from 4 to 47. Leaders gained awareness of zoning and the rights of tenants. Increased community empowerment and awareness on health impacts related to zoning, green spaces, and housing. Published a newsletter.	Yes
(14)	Fawcett 2018 [14]	United States	Empirical Case Study	Marginalised neighbourhoods/communities/groups	Healthcare access and equity; culturally appropriate healthcare access	Playbook for implementing organisational change	To help assure linkage to quality and culturally competent health services within the Latino Health for all coalitions	Organisational structures were established to address the intermediary aim of cultural competence in both health departments involved. More than 100 people received training in implementing the diabetes prevention program. Insurance enrolment numbers also increased during the intervention period, even though it could not be attributed to the intervention alone. The study provided a model for how health and community organisations can work together to improve health access of the vulnerable population groups.	Yes
(15)	Flick 1994 [15]	United States	Qualitative research	Marginalised neighbourhoods/communities/groups	Healthy eating; Healthcare access and equity; Other: Health literacy	Freir’s theory of adult education	To analyse efforts to empower a community through a partnership between a diverse, integrated neighbourhood in a large city and a graduate program for community health nurses.	In one case study, a school was prevented from closure. The second case study showed that the threat to health centre funding worsened conflict in the community organising group, created factions and destroyed trust. Funding was eventually secured but the funding conditions caused further conflict even though the funding was extended.	NA +
(16)	Haseda 2019 [38]	Japan	Quasi-experimental	Older adults	Older adults’ social activities	Japan Gerontological Evaluation Study (JAGES) Health Equity Assessment and Response Tool	To explore the effectiveness of community-organizing interventions on older adults’ participation in social activities	Local health staff member empowerment increased older male residents’ involvement in social activities. However, the frequency of going out did not show a clear association with the intervention.	Yes
(17)	Hatch 1978 [16]	United States	Case report	General population	Health education	Rothman’s locality development model	To develop problem-solving capacities and improve community integration through organising and assimilate information effectively to improve community health	Even after the original steering committee was discontinued, community members organised their own council. Funding was arranged for a housing program and a community leaders’ health education training program was successfully developed. The community members identified their new community leadership roles to come up with solutions	Yes
(18)	Hays 2003 [17]	United States	Process evaluation	People/communities of diverse genders (e.g., people who identify as non-binary or LBTQIA+)	Sexual/reproductive health and prevention	Alinsky-style organising (also known as IAF (Industrial Areas Foundation) style organising)	To build a strong, supportive young gay and bisexual men’s community where young gay and bisexual men nurture and protect each other, particularly from HIV	The Mpowerment Project mobilised youth to embrace their identities and support each other to take action on safer sex through different outreach events, and team performances such as “Gaywatch” bar outreach. Large events, such as dances, house parties, community forums, picnics, art shows were conducted as formal outreach. Smaller events such as weekly video parties, discussion groups and sports activities were also conducted. Beyond its preventive scope, the project creates a collective, community empowerment enabling the young men to organise together for taking on several other challenges they face.	Yes
(19)	Hedley 2002 [37]	Canada	Qualitative research	Older adults	Healthy eating	McKenzie & Smeltzer 1997 community organisation	To outline the development of a nutrition education program for older adults using a community organisation approach, including how it is revised based on evaluation and a community’s expressed needs.	The participants were more informed about nutrition and resources and started eating better (anecdotally) post intervention. They also enjoyed providing advice and opinions about appropriate nutrition activities	Yes
(20)	Hildebrandt 1994 [39]	South Africa	Project evaluation	Older adults	Health promotion (non-specific)	Eight step process	To meet community health needs through self-care and CIH, especially among vulnerable or disadvantaged populations.	Four programs were implemented—1. Health education & screening programs 2. Literacy program 3. Food gardening program 4. Nutrition education including food preparation and basic nutrition	Yes
(21)	Hilgendorf 2016 [18]	United States	Process evaluation of a pilot project	General population	Obesity	Collective Impact; Coalition Action	To summarise the lessons learned from a novel approach by the Wisconsin Obesity Prevention Initiative for community action towards obesity prevention	Lessons learnt from the project were: Developing understanding and capacity for coalition action and community organising takes time. Community organising helped local concerns related to the root causes of obesity, including poverty and transit resurface. Coalition and community organising also drew attention to cultural assets for health promotion (for example: traditional food practices, and links between cultural loss and obesity)	NA +
(22)	Kang 2015 [19]	United States	Qualitative research	Multiracial and historically marginalised communities	Healthy lifestyles (non-specific lifestyle prevention/modification)	Community-based participatory research (CBPR) and Psychosocial capacity building model	To build resilience and health in economically disadvantaged communities develop collective efficacy and social cohesion	Intergenerational community organising can be a good alternative to the youth-leadership development approach for social workers; Engaging in a healthy collective contributes to a sense of connection and affirmation; the community members became actors and creators of collective knowledge through the course of the study	Yes
(23)	Livingston 2018 [20]	United States	Community Intervention trial	First Nations (Cherokee) children and adolescents in Oklahoma	Substance use (alcohol, tobacco, illicit and includes violence and associated behaviours)	Communities Mobilizing for Change on Alcohol (CMCA)	To evaluate effects of 2 alcohol prevention interventions—a community organizing intervention designed to reduce youth alcohol access, and an individual-level screening and brief intervention to reduce other drug use outcomes	Both interventions were associated with statistically significant decreases in the number of nonalcohol drugs used in the past 30 days.	Yes
(24)	McKenzie 2004 [35]	UK	Quasi-experimental	General population	Cancer prevention	Community Organising and Building Model as outlined by McKenzie & Smeltzer (2001).	(a) To raise awareness about factors increasing an individual’s risk of developing cancer and ways to reduce the risk (b) To provide screening opportunities for colorectal cancer to individuals who are at higher risk but are likely to be missed out of the screening	A multiactivity intervention was created alongside two award-winning videos. 150 community volunteers were trained to deliver the education/screening program. 185 sessions reached more than 6000 individuals. The teachers of the intervention schools perceived the program to have been useful among their students. Cancer awareness was found to have been raised in the participants and more than 65% of the surveyed individuals said they would change at least one behaviour to reduce their cancer risk.	Yes
(25)	Parker 2010 [21]	United States	Qualitative research	Children and adolescents	Local environment factors (as a determinant of health/wellbeing); Other: Asthma	Community capacity framework	To identify the dimensions of community capacity that were enhanced as part of a CBPR community health development approach to reducing physical and social environmental triggers associated with childhood asthma and the factors that facilitated or inhibited the enhancement of community capacity	Several dimensions of community capacity were enhanced as part of a CBPR community health development approach, including in leadership, community participation, knowledge and policy advocacy skills, resource identification, social and organisational networks, a sense of community, understanding of community history, community power and values.	Yes
(26)	Perry 2000 [22]	United States	Community randomised trial	Children and adolescents	Substance use (alcohol, tobacco, illicit and includes violence and associated behaviours)	Community Action; Collective Impact	To describe the development and implementation process of Project Northland intervention which is focused on delaying on-set and reducing adolescent alcohol use using community-wide, multiyear, multiple interventions	At the end of Phase one that spanned 7 years, there were significant reductions in alcohol use among intervention students—a 20% reduction in past-month drinking and a 30% reduction in past-week drinking. By the 10th year, even after 2 years without a substantive intervention program, there were no significant differences between the intervention and reference groups. By the end of 11th grade, after 1 year of Phase two intervention activities, students in the intervention group drank less, but this was not statistically significant. However, among baseline nonusers, the difference between groups in past-week alcohol use was marginally significant (*p* < 0.07) at the end of the 11th grade, suggesting some impact from the 11th-grade intervention among these students.	Yes
(27)	Poole 1997 [23]	United States	Longitudinal study	Children and adolescents	Health promotion (non-specific); Healthcare access and equity	Collective Impact; Community Health Planning Committee	To improve children’s access to health care through Medicaid’s Early and Periodic Screening, Diagnosis, and Treatment (EPSDT) and other public health programs	The Medicaid officials accepted the data collected and reported by the Healthy Kids project as reliable. The screening ratios in the Garfield County where the community organising was implemented was found to be the highest and it was attributed to the Healthy Kids project.	Yes
(28)	Rask 2015 [24]	United States	Qualitative research	Health care facilities and organisations	Coordination of health services to prevent avoidable emergency presentations	Community organising process	To reach out to healthcare providers and organisations to determine initiatives for supporting reduced hospital readmissions.	There were reductions in readmissions to the Anchor hospital system from the participating facilities. Over the time, the participating facilities became more capable of retaining patients during the critical first 30 days after discharge.	Yes
(29)	Ross 2011 [25]	United States	Case report	Children and adolescents	Substance use	Community-based participatory research (CBPR)	To explore factors that facilitate and pose barriers to active youth involvement in a long-term, tobacco-related community change initiative.	The Teens Tackle Tobacco project exposed the young people’s sense of (in)justice. CBPR helped the youth develop research skills needed to channel their outrage into an effective and multilevel campaign. The project helped change their view of themselves as community change makers.	Yes
(30)	Salem 2005 [26]	United States	Descriptive research design	General population	Healthy lifestyles Healthy eating; Healthcare access and equity; Other: Community safety, economic development	Collective Impact	(a) To increase the capacity of communities to participate in public health decision making; (b) To promote new partnerships, both within individual communities and across the city; (c) To provide communities with an organised voice and access to decision makers; and (d) To determine an appropriate role for the public health department in supporting community-based health improvement activities	- Increased the availability of healthy foods as the proportion of neighbourhood stores selling fresh produce increased from 32% to 58% - Another effort aimed to increase community participation in crime-prevention activities was a success, with increased attendance at meetings in the two police beats where the block clubs were formed increased by 75 percent. - Development of new partnerships, such as a partnership between a local social service agency and school that resulted in grant funding to improve test scores and offer after-school programs; the availability of preventive health care screenings and exercise classes at block parties and other community forums - Community organizations and residents have learned how to work as a team through community initiatives such as the garden, block clubs, and community meetings	Yes
(31)	Santilli 2016 [27]	United States	Longitudinal study	General population	Health promotion (non-specific); Healthy lifestyles (non-specific lifestyle prevention/modification); Better wellbeing	Community organising principles	To strengthen capacity, responsibility, and sustainability of neighbourhood-driven health interventions focused on chronic disease prevention and healthy lifestyles	42% of survey participants reported changes in their neighbourhood that make living a healthier lifestyle easier. Fewer participants reported eating unhealthy foods and getting no exercise. The project results were also used by neighbourhood associations, community-based and non-profit organizations, hospitals and health centres, governmental entities, and businesses in presentations, health services planning, reports, grant applications, and business plans.	Yes
(32)	Saxon 2021 [28]	United States	Case report	Women	Gender discrimination/violence/hostility; Healthcare access and equity	Ganz’s community-organizing model	To understand the processes to facilitate change and improve health among underserved populations in three programs in Sri Lanka and Bangladesh	The study found that public health approaches can benefit from community organizing to develop local engagement and participation. The study concluded that a data-driven public health approach combined with community-based participatory efforts contributes to this change.	Yes
(33)	Subica 2016 [29]	United States	Project evaluation	Children and adolescents	Obesity; Healthy lifestyles Physical activity; Healthy eating; Local environment factors	Alinsky-style organising (also known as IAF (Industrial Areas Foundation) style organising)	To presents outcomes of the Robert Wood Johnson Foundation’s Communities Creating Healthy Environments (CCHE) initiative: the first national program to apply community organizing to combat childhood obesity-causing structural inequities in communities of colour	The CCHE grantees utilised community-organising practices to generate 72 environmental and policy solutions to childhood obesity across 21 communities, across six domains: two directly addressed childhood obesity by enhancing children’s healthy food and recreational access, whereas four indirectly addressed obesity by promoting access to quality health care, clean environments, affordable housing, and discrimination- and crime-free neighbourhoods.	Yes
(34)	Tataw 2020 [30]	United States	Case study design	Community members living in a health district comprised of four-county region	Reporting on the state of community partnership as a part of Community Health Improvement Plan implementation	A conceptual framework that combines organizing theory and horizontal participatory approaches	(a) To use a conceptual framework to guide both process analysis and survey data analysis (b) To identify strengths and limitations in partnership and participation in the planning and implementation of a community health improvement plan (c) To discuss the implications for community health research and practice	Partner synergy, cultural and structural relationships were found to be strong in the horizontal participatory approach, while community participation (especially of the minorities and marginalised groups) and social action were found minimal. A complete participatory model of practice promoting a wider community participation was recommended.	NA +
(35)	Wagenaar 1999 [31]	United States	A 15-community randomised trial	General population	Substance use (alcohol, tobacco, illicit and includes violence and associated behaviours)	Communities Mobilizing for Change on Alcohol ‘CMCA’ to implement changes in local institutional policies.	To describe actions taken by organisers and strategy teams across the seven intervention communities and share lessons and results of implementing the large-scale community trial	The intervention teams changed policies and practices of community institutions such as law enforcement agencies, alcohol merchants, and sponsors of community events. This led to significant changes in alcohol-related behaviours among 18- to 20-year-olds, and significant reductions in the alcohol establishments’ tendency to serve alcohol to youth.	Yes
(36)	Wagoner 2010 [32]	United States	Cohort study	Young adults	Substance use (alcohol, tobacco, illicit and includes violence and associated behaviours)	Community organising conceptual model	(a) To explore and characterise the process of community organising used by university-based community organisers. (b) To develop and confirm the use of a community organizing model to create coalitions and implement environmental change on college campuses and surrounding communities.	Severe consequences due to students’ alcohol consumption and alcohol-related injuries were reduced after 3 years of intervention.	Yes
(37)	Weeks 2013 [33]	United States	Mixed methods observational case study design	“At-risk” population and their peers, health and social service patients and clients.	Sexual/reproductive health and prevention	Community empowerment, social ecology, social learning, innovation diffusion.	(a) To develop a coalition that is able to design and implement and sustain multilevel interventions to increase availability, accessibility and support for female condoms (b) To design and implement a multilevel intervention to increase availability, accessibility and support for female condoms	A community coalition was developed, and the interventions developed were received positively by the target population. The use of female condom improved, shops began introducing female condoms, there was an increased promotion of the product. The authors identified these changes suggesting potential for sustainable changes at individual, organisational and community levels.	Yes
(38)	Zanoni 2011 [34]	United States	Case study	Children and adolescents	Obesity; Other: Asthma	Community organising with a focus on power	To confront the epidemics of asthma and obesity in Latina/o children that result in premature death through an environmental justice partnership between health researchers and Latina/or organisations	The study found that community-based health partnerships help parents raise their voices and challenge the status quo of social inequity by taking actions to affect their children’s schools and community.	Yes

* Intervention success defined as per the original study’s findings. + Not Available.

**Table 2 ijerph-20-05341-t002:** Core components of frameworks/model/process and details of selected studies (n = 38).

S. No.	Author/Year	Framework Used	Core Components of the Framework	Framework/Model/Process	Power
1	Agrusti 2020 [2]	Community needs assessment methodology model	Nine steps community based participatory approach was created: (1) Identify assessment teams and roles, communities of focus, (2) Review past assessment findings and set aims, (3) Establish study methodology, (4) Collaborate in data collection, (5) Conduct thematic and statistical analysis, (6) Present preliminary findings to stakeholders, (7) Compare results, (8) Prepare reports, (9) Disseminate findings. Throughout the process, stakeholders were engaged in planning, assessment and dissemination. University research team integrated with the government customer and empowered the stakeholders to engage in key decisions.	Model	N/A
2	Bauermeister 2017 [3]	Community-based participatory research	Engage researchers and community partners through shared decision making. This community engagement approach offered an alternative to traditional research by challenging the notion of “researcher-as expert” and centring community expertise and lived experience. Use community dialogues in 1. round tables (community listening) 2. Refinement 3. Prioritisation. The community dialogue process used in this article is consistent with the principles of CBPR, and it helped the authors to ensure that the program was tailored to the specific needs of the community and that it was well-accepted by the community. However, more information about the recruitment process and measures to ensure participants’ confidentiality and safety would have been beneficial.	Framework	Shared power through collaboration with community and researchers. Each minority group e.g., transgender only and youth-only Gives each group a chance to participate in an open space without being intimidated or silenced by older community members and/or professionals. Also considered geographic diversity
3	Berman 2018 [4]	Implementation science framework (Proctor and colleagues)	Healthy Lifestyles Initiative used five implementation strategies to support organizations in using the messaging materials and implementing policy, systems, and environmental activities: (1) educational training, (2) a structured action plan, (3) coalition support, (4) one-on-one support, and (5) materials dissemination and resource sharing.	Framework	N/A
4	Bezboruah 2013 [5]	Community organising conceptual framework	The study adopted a community organising approach where semi structured interviews with executives of several non-profit organisations and community organisers were collected, also participated in events and meetings organised for community organising. Collected da Data collected from the interviews are analysed in a systematic manner that assisted in theory building	Framework	N/A
5	Black 2020 [36]	Green care theory	The study conducted the collaborative performance, working with the garden users and worked in the garden. It was focused on connectedness to the natural environment. Recorded observations of actions and interactions as fieldnotes. Interviews were conducted and recorded with three volunteers and staffs. Also recorded the personal observations and impressions in person.	Framework	N/A
6	Bosma 2005 [6]	Community organizing model	The community organizing component consisted of five stages: (1) Assessment, (2) Action team creation, (3) Creation of an action plan, (4) Mobilization and action, and (5) Implementation	Model	Community members (youth and adults) were empowered to address issues relating to alcohol, tobacco, marijuana, and violence
7	Brookes 2010 [7]	Community organisation and community partnerships	The program was combination of community organising and collaboration. Implemented with five components: Community partnership, parents education programs, major events, online events and a media campaign. For the community partnership, coalition between different partner groups and parents were created. Additionally, community influentials, leading professional, mayors were included in advisory committee. Parents met on regular basis to develop educational programs and events.	Process	N/A
8	Bryant 2010 [8]	Innovative program planning framework, community-based prevention marketing	9 steps approach: 9-step process: (1) mobilise the community; (2) develop a profile of community problems and assets; (3) select target behaviours, audiences, and when possible, interventions to tailor; (4) build community capacity to address the priority or target problem; (5) conduct formative research; (6) develop a marketing strategy; (7) develop or tailor program materials and tactics; (8) implement the new or tailored intervention; and (9) track and evaluate the program’s impact	Framework	Capacity building activities such as the marketing skills and participatory research techniques for designing, tailoring, and implementing interventions that promote behaviour change were provided to the interested members.
9	Cheadle 2009 [9]	Community organising model	The SESPAN project was implemented on the basis of community organising approach where community organisers were hired to develop partnerships and network among community-based organisations, groups and institutions. These community organisations were focused on physical activity. Relationships between the key organisations were built through coalition and one-on-one networking. Semi structured interviews with community stakeholders, a variety of survey-based measures of older adults including pre/post survey were included.	Process	Partnered with many local organizations and sustain SESPAN activities after the 5-year research funding period ends.
10	Cheadle 2010 [10]	Social ecologic model	Micro, meso and macro level input- combining individual-level programs with larger scale environmental and policy change follows the social ecologic model. The authors used a mixed-methods approach to evaluate the effectiveness of the community-organizing approach. They collected data on changes in physical activity, diet, and body weight through surveys, focus groups, and objective measures. They also collected data on the acceptability and feasibility of the interventions through focus groups and interviews with community members and organizations. Overall, the article describes a community-organizing approach that is grounded in CBPR principles and uses a variety of methods to engage community members in the planning and implementation of interventions to promote physical activity.	Framework	Involvement in coalitions
11	Denham 1998 [11]	Cottrell’s Community Competency framework	Cottrell’s eight dimensions for community to function as a collectively included: (1) Commitment, (2) Self-other awareness and clarity of situational definitions, (3) Articulateness, (4) Communication, (5) Conflict containment and accommodation, (6) Participation, (7) Management of relations with the larger society, and (8) Machinery for facilitating participant	Framework	N/A
12	Doherty 2006 [12]	Citizen Health Care	Core principles of Citizen Health Care model are: (1) The greatest untapped resource for improving healthcare is the knowledge, wisdom, and energy of individuals, families, and communities who face challenging health issues in their everyday lives. (2) People must be engaged as coproducers of healthcare for themselves and their communities, not merely as patients or consumers of services. (3) Professionals can play a catalytic role in fostering citizen initiatives when they develop their public skills as citizen professionals in groups with flattened hierarchies. (4) If you begin with an established program, you will not end up with an initiative that is “owned and operated” by citizens, but a citizen initiative might create or adopt a program as one of its activities. (5) Local communities must retrieve their own historical, cultural, and religious traditions of health and healing and bring these into dialogue with contemporary medical systems. (6) Citizen health initiatives should have a bold vision (a BHAG. a big, hairy, audacious goal) while working pragmatically on focused, specific projects.	Model	N/A
13	Douglas 2016 [13]	Communities Creating Healthy Environments (CCHE) Change Model and Evaluation Frame	Consist of five overarching strategies grounded in individual, organizational, and community empowerment processes and outcomes: (1) Developing a community base sympathetic to, and supportive of, public health change initiatives. (2) Building leader base, (3) Building ally base, supported by an aligned base of organizational allies with shared interests and values poised to work together toward community health equity, (4) Message reframing and (5) Activate and maintain ongoing community base participation in public health initiatives.	Framework	Empowering communities to directly redress health inequities
14	Fawcett 2018 [14]	Playbook for implementing organisational change	(A) Playbook for implementing organizational change for cultural competence: (1) initial orientation and commitment to engage, (2) assessment of the current organization or program, (3) dialogue on identified gaps and priority setting, (4) action planning: draft created by the smaller team and whole group review, (5) implementation and monitoring of progress, and (6) closing dialogue and celebration of achievements (B) playbook for improving quality through access to preventive health services and the Diabetes Prevention Program: (1) initial brief orientation session with potential partners, (2) review of recommendations and plan development, (3) pilot test of implementation protocol to identify prediabetic clients and referral protocols, (4) implementation and monitoring of progress, and (5) dialogue and celebration of achievements. (C) Playbook for improving access and linkage to care through insurance enrolment. The participatory process used five elements (1) initial orientation and dialogue about partnering, (2) review/commit to a level of partnership and related responsibilities, (3) development of an action plan, (4) implementation (typically, during the ACA enrolment period), and (5) monitoring and evaluation.	Process	The Coalition successfully engaged Latinos and other marginalised groups by partnering to enhance access and linkage to quality health services
15	Flick 1994 [15]	Freire’s theory of adult education	Community mobilisation occurs through community participation and control. The professional serves as a resource and catalyst but program ideas and direction come from the community. Partnerships with the communities were made based on reciprocity, trust developed through continuous long-term involvement, social justice with its inherent assumption of equity, and a broad definition of health that includes well-being and a sense of community. Faculty was continuously involved to understand the interpersonal and political relationships among community residents and organisations and to identify when action was taken regarding an existing problem.	Model	Empowered the community as a whole and increase its capacity to improve its own health.
16	Haseda 2019 [38]	JAGES Health Equity Assessment and Response Tool	The research team visualised and figured out community health needs by using a JAGES Health Equity Assessment and Response Tool and developed community diagnosis forms. Municipality health sector staff members were supported to utilise the community assessment data tool (JAGES-HEART) and promote intersectoral collaboration, aiming to develop health-promoting social activities in the community.	Framework	Researchers empowered local health sector staff members
17	Hatch 1978 [16]	Rothman’s locality development model	Community diagnosis was carried out to identify community needs and methods for utilising resources available inside and outside of the community to meet these needs. Field team members and students continue their responsibilities as requested by the committee and became involved in publicity and media reporting for the program. Each committee member functioned as a group leader assuring the completion of his task at the designated time and shared equal responsibilities. Team members continued attending meetings and talking with more community people and recognised and supported the outstanding work of committee members.	Model	N/A
18	Hays 2003 [17]	Alinsky-style organising (also known as IAF style organising)	Hiring coordinators, 2. community assessment 3. community organising through core groups, goal development, Action 4. Linking with another organisation 5. Tailoring project to community. The Mpowerment Project is a community-based intervention that uses a combination of community organising, peer education, and social marketing to empower young gay and bisexual men to take control of their health and reduce their risk of HIV infection. The authors used a CBPR approach and a variety of methods to engage and evaluate the young gay and bisexual men in the project.	Framework	Shared power through involvement of core group who represent the various segments of the target community
19	Hedley 2002 [37]	McKenzie & Smeltzer 1997 community organisation	The eight steps of community organising model were adapted to develop the program which includes, (1) Citizens recognise the problem, (2) Organise the people, (3) Identify specific problems, (4) Set goals and establish priorities, (5) Choose solutions or activities, (6) Implement the action plan, (7) Evaluate the plan and process, (8) Modify and expand the plan	Model	Members took on greater responsibility on leading the implementation and evaluation of the program
20	Hildebrandt 1994 [39]	Eight step process	The initial steps were focused on community contacts and one-to-one relationships with individuals and groups to identify barriers and assets in the community. The intermediate steps facilitated the open meeting to identify the community needs and their solutions. The final steps were focused on maintaining and sustaining the program. Eight steps of the models were: (1) Information seeking, (2) Support seeking, (3) Set up a work group and a plan with goals, (4) Identify tasks with deadlines and person responsible for each, (5) Interim deadlines and startup date, (6) Nurture the new program, (7) Measure against the original goals, (8) Keep the community informed of progress	Model	N/A
21	Hilgendorf 2016 [18]	Collective Impact; Other: Coalition Action	5 Dimensions of collective impact. 1. Backbone organisation 2. Common agenda 3. shared data platform 4. shared vision 5. Communication	Framework	The organisation supports broad participation of residents in the democratic process, especially through congregation-based community organizing
22	Kang 2015 [19]	Community-based participatory research (CBPR)	An alternative paradigm of knowledge production in which groups who are adversely affected by a social problem undertake collective study to understand and address it. The author implemented CBPR by closely involving the community members in the research process, prioritizing community ownership, and addressing potential biases by inviting multiple perspectives and using techniques such as member-checking and negative peer analysis throughout the data analysis process.	Framework	Shared power through collaboration with the community and researchers-uses an intergenerational approach
23	Livingston 2018 [20]	Communities Mobilizing for Change on Alcohol (CMCA)	An iterative process with six stages of community organising was adopted to implement the intervention CMCA, (a) assessment of community interests through face-to-face, one-on-one or two-on-one meetings with hundreds of community residents; (b) building a base of support through one-on-ones and establishment of a community action team; (c) expanding the base of support through one- or two-on-one meetings, presence and presentations at community events, and media advocacy; (d) development of a plan of action; (e) implementation of actions; and (f) maintenance of effort and institutionalisation of change.	Model	N/A
24	McKenzie 2004 [35]	Community Organizing and Building Model as outlined by McKenzie & Smeltzer (2001).	Mckenzie and Smeltzer used a model with 10 steps to bring behaviour change in the targeted population. Those steps were: 1. Recognizing the concern, 2. Gaining entry into the community, 3. Organizing the people, 4. Assessing the community, 5. Determining the priorities and setting the goals, 6. Arriving at a solution and selecting an intervention, 7. Implementing the plan, 8. Evaluating the outcomes of the plan of action, 9. Maintaining the outcomes in the community, 10. Looping back	Model	N/A
25	Parker 2010 [21]	Community capacity framework (Freudenberg 2004, Goodman et al. 1998)	Community organisers were hired to work with community groups. Interviews with key stakeholders were conducted to prioritise major areas upon which to focus their community capacity-building efforts. Education and data to community members and policy makers were provided to understand the potential health implications of the proposed projects.	Model	Authors has discussed about capacity building among community members.
26	Perry 2000 [22]	Collective Impact; Community Action	Behavioural model to change community efficacy and norms through market and policy levers. Use community organising process. Community listening, community action teams, Action plans, execution of action plan in community	Process	The members of the teams were a small percentage of the entire intervention cohort, and so this direct empowerment opportunity was not experienced by most of the cohort. The purpose of these interviews was to identify each community’s social, economic, and power structures; determine both the community’s and the interviewee’s interest in reducing high school students’ access to alcohol; determine how the problem was perceived in the community; and build a broad base of support for future actions.
27	Poole 1997 [23]	Collective Impact; Other: Community Health Planning Committee	Community organising-not collective impact- Process: Action structures, Community Problem-Solving Process,	Process	To ensure that solving local problems is a shared responsibility, all Metro Commission projects are community partnerships. This reflects the organization’s philosophy that local needs are community owned, and that meeting them is a shared responsibility, not the responsibility of any one sector or service entity.
28	Rask 2015 [24]	Community organising process	The community organising process consists of five phases: community assessment, coalition building, strategic planning, action, and sustainability.	Framework	N/A
29	Ross 2011 [25]	Community-based participatory research	PYD and SJYD are used in the context of CBPR to ensure that the youth are actively involved in the initiative, that their needs and perspectives are taken into account, and that the initiative promotes social justice and addresses health disparities among marginalised youth. By using PYD and SJYD in CBPR, the initiative is able to create a sense of ownership, investment, and empowerment among the youth while also addressing the social and economic determinants of health and reducing health disparities among marginalised youth. The article concludes that PYD and SJYD can be an effective approach for engaging marginalised youth in long-term tobacco control initiatives.	Framework	Shared power. The youth’s research and action on this issue inspired key decision makers, including a city councillor, the director of the city’s Tobacco Control Program, the city’s director of Public Health, and a state senator to embrace the youth’s cause by developing new policies and ordinances.
30	Salem 2005 [26]	MAPP model	Mobilizing for Action through Planning and Partnerships (MAPP). 1. Organise for success- partnership development, 2. Visioning 3. Four MAPP assessments 4. Identify strategic issues 5. Formulate goals and strategies, 6. Action (plan, implement, evaluate)	Framework	Develop coalitions-developing community voices by working with the alderman to ensure that community residents are able to share their concerns
31	Santilli 2016 [27]	Community organizing principles	Worked closely with community residents and community-based organisations to develop trust partnerships and to gain deep knowledge of history, norms and leadership. Before the survey, community organisers were hired, letters were emailed, conducted one-on-one meetings with community members and based on a well-established relationship with the local press, a press conference was held. Community youth volunteers were trained for data collection and survey methods and analysed data were disseminated to the community.	Process	N/A
32	Saxon 2021 [28]	Ganz’s community-organizing model	In the study, community organizing was narrow down to four central components, (1) Building relationships, (2) Telling the story, (3) Devising strategy and (4) Catalysing action	Model	Research has shown that participating in community organizing tends to give people more ownership over local issues.
33	Subica 2016 [29]	Alinsky-style organising (also known as IAF style organising) Community organizing–based health promotion consists of grassroots movements (interventions) that raise individuals’ collective capacity to control their social and built environments by advocating for public policies that balance decision-making power and resource distribution toward health equity	Community grants, the article describes a community organizing approach that is grounded in CBPR principles and uses a variety of methods to engage community members in the identification, prioritization, and addressing of health issues. The study used a mixed-methods approach to evaluate the effectiveness of the approach and gather feedback from community members on acceptability.	Framework	N/A
34	Tataw 2020 [30]	Conceptual framework that combines organizing theory and horizontal participatory approaches	The program used the integrated framework of explanatory, change and organising theories for the community health improvement plan life cycle in three stages: health problems clarification; organising; issue prioritisation, and program activities	Framework	N/A
35	Wagenaar 1999 [31]	Community organizing approach	Intervention community followed an organizing process that included seven stages, 1. Assessing the community, 2. Creating a core leadership group, 3. Developing a plan of action, 4. Building a mass base of support, 5. Implementing the action plan, 6. Maintaining the organization and institutionalizing change, 7. Evaluating changes	Framework	Used power mapping for the data collection, became familiar with the demographics of their communities, the power relationships within the community.
36	Wagoner 2010 [32]	Community organising conceptual model	A full-time community organiser was hired who was familiar with substance abuse prevention, knowledge of environmental approaches to health behaviour change, and had experience in community organising. In-depth interviews of an average 60 min were conducted among community organising members. All interviews were audio-recorded and non-verbal reactions were recorded by a note taker. Data were analysed and presented.	Model	Assessed both the problem of alcohol use and the power dynamics of their campus
37	Weeks 2013 [33]	Community empowerment, social ecology, social learning, innovation diffusion.	Community engagement approach using various theories. Community action advisory board (CAAB), CAAB mobilisation and capacity building. It used a multiple case study design, which allowed the authors to gain a more in-depth understanding of the issues related to creating CAABs in different contexts. The use of both interviews and document review also allowed the authors to triangulate data and strengthen the reliability of the findings.	Framework	Authors approached the CAAB training recognizing that members were grassroots “experts” and “leaders” in key areas of HIV/STIs prevention, women’s health and empowerment and community health needs.
38	Zanoni 2011 [34]	Community organising with a focus on power	Little Village Environmental Justice Organization (LVEJO) hired an organiser (parents with children in community school) to communicate and lead the discussion on youth obesity and overweight with community participants through outreach activities. Parents received the training for the semi structured interview, documenting and reporting. Based on the results action plan was developed	Process	One of the parents/teachers internalised the risk of obesity in her daughter after looking at the program module and developed many activities for her students to prevent obesity. Parents are the main persons who will observe and change their children’s habits

**Table 3 ijerph-20-05341-t003:** Mixed Methods Appraisal Tool (Version 2018) for critically appraising quantitative (n = 14), qualitative (n = 16), and mixed methods (n = 2) study reviews.

First Author (Year)	Question						
Qualitative	QS1	QS2	Q1.1	Q1.2	Q1.3	Q1.4	Q1.5
Bezboruah 2013 [5]	Y	Y	Y	Y	Y	Y	Y
Black 2020 [36]	Y	Y	Y	Y	Y	Y	Y
Denham 1998 [11]	Y	Y	Y	Y	Y	Y	Y
Doherty 2006 [12]	Y	Y	Y	U	Y	Y	U
Douglas 2016 [13]	Y	Y	Y	Y	Y	Y	Y
Flick 1994 [15]	Y	Y	Y	Y	Y	Y	Y
Hatch 1978 [16]	Y	Y	Y	Y	Y	Y	Y
Hedley 2002 [37]	Y	Y	Y	Y	Y	Y	Y
Hilgendorf 2016 [18]	Y	Y	Y	Y	Y	Y	Y
Kang 2015 [19]	Y	Y	Y	Y	Y	Y	Y
Parker 2010 [21]	Y	Y	Y	Y	Y	Y	Y
Ross 2011	Y	Y	Y	Y	U	Y	Y
Saxon 2021 [28]	Y	Y	Y	Y	Y	Y	Y
Tataw 2020 [30]	Y	Y	Y	N	Y	N	N
Wagoner 2010 [32]	Y	Y	Y	Y	Y	Y	Y
Zanoni 2011 [34]	Y	Y	Y	Y	Y	Y	Y
Quantitative (RCT)	QS1	QS2	Q2.1	Q2.2	Q2.3	Q2.4	Q2.5
Livingstone 2018	Y	Y	Y	Y	Y	N	U
Perry 2000 [22]	Y	Y	Y	Y	N	N	Y
Wagenaar 1999 [31]	Y	Y	U	Y	Y	N	Y
Quantitative (Non-randomised)	QS1	QS2	Q3.1	Q3.2	Q3.3	Q3.4	Q3.5
Bosma 2005 [6]	Y	Y	Y	Y	Y	N	Y
Brookes 2010 [7]	Y	Y	Y	Y	Y	N	Y
Byrant 2010	Y	Y	Y	Y	Y	Y	Y
Cheadle 2010 [10]	Y	Y	Y	Y	U	U	Y
Haseda 2019 [38]	Y	Y	Y	Y	Y	Y	Y
McKenzie 2004 [35]	Y	Y	Y	Y	Y	Y	N
Subica 2016 [29]	Y	Y	Y	U	Y	N	Y
Quantitative (Descriptive)	QS1	QS2	Q4.1	Q4.2	Q4.3	Q4.4	Q4.5
Bauermeister 2017 [3]	Y	Y	Y	Y	Y	U	Y
Berman 2018 [4]	Y	Y	Y	Y	Y	N	Y
Poole 1997 [23]	*Y*	*Y*	*Y*	*Y*	*Y*	*U*	*U*
Salem 2005 [26]	Y	Y	Y	Y	Y	N	Y
Mixed Methods	QS1	QS2	Q5.1	Q5.2	Q5.3	Q5.4	Q5.5
Agrusti 2020 [2]	Y	Y	Y	Y	Y	Y	Y
Weeks 2013 [33]	Y	Y	Y	Y	Y	Y	Y
Further appraisal is not feasible							
Cheadle 2009 [9]	U	Y					
Fawcett 2018 [14]	Y	N					
Hays 2003 [17]	Y	N					
Hildebrandt 1994 [39]	Y	N					
Rask 2015 [24]	N	Y					
Santilli 2016 [27]	N	N					

Index Abbreviations: Y = Yes, N = No, U = Unclear, MMAT = Mixed Methods Appraisal Tool. A. Mixed Methods Appraisal Tool (MMAT) screening questions for all studies—QS1: Are the research questions clear?, QS2: Do the collected data allow to address the research questions? B. Mixed Methods Appraisal Tool (MMAT) for qualitative studies—Q1.1: Is the qualitative approach appropriate to answer the research questions?, Q1.2: Are the qualitative data collection methods adequate to address the research question? Q1.3: Are the findings adequately derived from the data?, Q1.4: Is the interpretation of results sufficiently substantiated by data?, Q1.5: Is there coherence between qualitative data sources, collection, analysis and interpretation? C. Mixed Methods Appraisal Tool (MMAT) for quantitative Randomised Controlled Trail—2.1. Is randomization appropriately performed? 2.2. Are the groups comparable at baseline? 2.3. Are there complete outcome data? 2.4. Are outcome assessors blinded to the intervention provided? 2.5 Did the participants adhere to the assigned intervention? D. Mixed Methods Appraisal Tool (MMAT) for quantitative non-Randomised—3.1. Are the participants representative of the target population? 3.2. Are measurements appropriate regarding both the outcome and intervention (or exposure)? 3.3. Are there complete outcome data? 3.4. Are the confounders accounted for in the design and analysis? 3.5. During the study period, is the intervention administered (or exposure occurred) as intended? E. Mixed Methods Appraisal Tool (MMAT) for quantitative descriptive studies—Q4.1: Is the sampling strategy relevant to address the research question?, Q4.2: Is the sample representative of the target population?, Q4.3: Are the measurements appropriate?, Q4.4: Is the risk of nonresponse bias low?, Q4.5: Is the statistical analysis appropriate to answer the research question? F. Mixed Methods Appraisal Tool (MMAT) for mixed-method studies—Q5.1: Is there an adequate rationale for using a mixed methods design to address the research question?, Q5.2: Are the different components of the study effectively integrated to answer the research question?, Q5.3: Are the outputs of the integration of qualitative and quantitative components adequately interpreted?, Q5.4: Are divergences and inconsistencies between quantitative and qualitative results adequately addressed?, Q5.5: Do the different components of the study adhere to the quality criteria of each tradition of the methods involved?

## Data Availability

Available on request.

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
