# Peer review of "Community Organising Frameworks, Models, and Processes to Improve Health: A Systematic Scoping Review"

_ijerph, 2023, doi:10.3390/ijerph20075341_

Round 1
Reviewer 1 Report
This well written paper will be of interest to all striving to improve population health. The authors should address 2 small points
1. The numbrs in figure 1 and section 8 do not match. Please review and reconcile
2. In the discussion please consider if there has been any trend / shift in use of methods / frameworks over time as this literature review spanned work published over a considerable time period.
Author Response
Comments and Suggestions for Authors
This well written paper will be of interest to all striving to improve population health. The authors should address 2 small points
Comment 1: The numbers in figure 1 and section 8 do not match. Please review and reconcile
Response to comment 1: It was due to a formatting issue. It has now been resolved.
Comment 2: In the discussion please consider if there has been any trend / shift in use of methods / frameworks over time as this literature review spanned work published over a considerable time period.
Response to comment 2: We have added two sentences on the trends in use of frameworks over time. Despite the use of community organizing frameworks over several decades, there is still no single gold standard framework adopted. A wide variety of models, frameworks or processes of community organising were applied in the included studies.
Reviewer 2 Report
Community Organising Frameworks, Models and Processes to Improve Health: A Systematic Scoping Review
This review documented the effectiveness of community involvement on influencing health behaviors/outcomes. This is a timely review. Community involvement is a significant investment of the community residents and the project leads, so an evaluation of outcomes elucidates how to optimize these approaches, as well as reinforcing the importance of setting realistic goals for achievable outcomes. Several comments and suggestions are provided below. Some are simple points of interest, and could perhaps be emphasized, and others are recommended revisions.
Abstract
See comment below in Introduction about: “Community involvement engages, empowers, and mobilises people to achieve their shared goals by addressing structural inequalities in the social and built environment.”
Appreciate that this is focused on effectiveness of interventions. Might discuss what an ideal state would look like? Five years after projects like these concluded, what might “effectiveness” look like? Do these provide lasting solutions or are they more transient (though still valuable to whomever was able to benefit).
The word “diversity” should be replaced here and later to avoid confusion with the ‘Diversity-Equity-Inclusion’ connotation. Perhaps simply repeat use of word “variation.”: “A wide variation was noted in the models, frameworks or processes of community organising utilised in these studies. We concluded that diversity implies that no single model, framework or process seems to have predominance over others in implementing community organising as a vehicle of positive social change within the health domain.”
Introduction
This first sentence (“Community involvement in research…”) should be scrutinized further, since this is the hypothesis that the paper is testing. I would not state this with such certainty and remind readers that these are the supposed/alleged/assumed implications of community involvement. For example, are their shared goals of health improvement being met after these projects are concluded? The term “empower” is ambiguous here.
Results
Is it worth reiterating that only 38 papers were found? I am glad the authors found that many, but it seems like a low number by any standard, especially with a beginning pool of nearly five thousand?
Also, I wonder how the sample size would change if the community involvement outcomes were based on environmental (e.g., air/water/soil quality) interventions. I suspect the literature would have more examples given the propensity of citizen/community science initiatives, but I also wonder if the results of the evaluation would be similar (I suspect they might).
This, I think, should be sure to be reiterated as a main finding: “The reviewed studies did not document long-term outcomes or seem to document measurable changes in health outcomes.” Because it does beg the question of the initial premise of “achieving shared goals” – i.e., do they mean during the project, or the long-term afterward?
It’s okay and highly relevant to scrutinize the effectiveness of these efforts, since these efforts represent significant investments in time and resources from both communities and researchers, so looking for best practices requires a critical eye.
Table 1
Provide more information inn table caption; define each column header. I assume “S. No.” is “sample number” but what is the second number before author’s name? Also caveat/define “Intervention success,” and that this is based on the original study’s findings, and they were not judged by this review (also explain the ‘NA’ designation; did they not try to assess it, or were they inconclusive?)
Scrutiny of the original evaluations could be emphasized. For example, “building trust and generating commitment” (e.g., #4) is fairly subjective.
This is an incredible finding, again because of the time/resource commitment. One must question whether this is an appropriate timeframe. I realize this is leading up to the “action phase” but this is still a long time and seems accurate based on my experience: “Most studies took 24 months or more to start the action phase.”
No Health Impact Assessments? (e.g., from PEW): https://www.pewtrusts.org/en/research-and-analysis/data-visualizations/2015/hia-map?sortBy=relevance&sortOrder=asc&page=1)
So all the studies used steps, and about half of them followed an existing framework. (The review identified 22 different frameworks, models, or processes adopted. There were 10 studies that explicitly reported the use of a community organising framework. All studies reported their use of community organising steps, and these varied between four and 10 steps.)
Table 2
Define “Relational Power”
Somewhere, take a sentence or two to define “Framework/Model/Process” – this is not critical and does not need to be over-precise, but it would help to have a basic frame of reference.
Noted that: “…only two studies evaluated their achievements or goals at the end of the program.”
8.1.3. CBPR
Mentioned above: **Wonder how results would have changed if ‘environmental’ was included in search parameters? Perhaps the sole focus on health is what yielded a seemingly low sample size (here and overall).
Much about “power” and I believe this is typically defined, but could use as example for evaluation challenges (Kang [27] and Bauermeister[13] highlighted the shared power. Ross et al. [33] described the power of youth’s research and participation/action on health behaviour)
8.2 Targeted health behaviors
Again, on the topic of “short term” evaluations: “The outcomes of most initiatives were promising, with positive changes reported (at least in the short term) in health outcomes for the target populations in most studies (32/38).”
Quality appraisal (this needs a heading number)
Discussion
Diversity – of options for community organizing
Change “Diversity” (mentioned above): A wide variety of models…. The diversity implies that no one specific model, framework or process seems to have predominance over others in implementing community organising
I would debate whether the “consistent themes” are so different (“…some consistent themes were prevalent across the reviewed studies…”)? These themes are very similar and subjective. If nothing else, perhaps an acknowledgment of this observation, or at least that many of these are relative/subjective to whomever is defining it.
Interesting: “However, this review suggested that community organising as a vehicle of health initiatives or interventions has yet to pick up traction in countries outside of the United States.”
Change “Diversity” (mentioned above): “Despite the heterogeneity in the selection of target population groups, there was consistency among most studies in terms of positive change reported in their targeted health outcomes. Such consistency in positive outcomes despite the diversity among target population groups reinforces the argument that community organising has the potential to be an important vehicle for positive change [12,13,20,25,27].”
Perhaps a table that summarizes primary conclusions from Discussion and Limitations? (e.g., While most studies reviewed reported positive change in the health outcomes, we noted that these measures were typically collected over relatively short and focused project durations. Many studies did not discuss long term, sustainability of positive impacts, particularly after the funding had been exhausted.)
For example, if they had worked to change an existing program, policy, or decision (that affects health), would that have longer term impacts?
Author Response
Community Organising Frameworks, Models and Processes to Improve Health: A Systematic Scoping Review
This review documented the effectiveness of community involvement on influencing health behaviors/outcomes. This is a timely review. Community involvement is a significant investment of the community residents and the project leads, so an evaluation of outcomes elucidates how to optimize these approaches, as well as reinforcing the importance of setting realistic goals for achievable outcomes. Several comments and suggestions are provided below. Some are simple points of interest, and could perhaps be emphasized, and others are recommended revisions.
Abstract
Comment 1: See comment below in Introduction about: “Community involvement engages, empowers, and mobilises people to achieve their shared goals by addressing structural inequalities in the social and built environment.”
Response to comment 1: Thank you, addressed as outlined below.
Comment 2: Appreciate that this is focused on effectiveness of interventions. Might discuss what an ideal state would look like? Five years after projects like these concluded, what might “effectiveness” look like? Do these provide lasting solutions or are they more transient (though still valuable to whomever was able to benefit).
Response to comment 2: Thank you for this suggestion, due to the long length of the paper already, we have decided not to include this.
Comment 3: The word “diversity” should be replaced here and later to avoid confusion with the ‘Diversity-Equity-Inclusion’ connotation. Perhaps simply repeat use of word “variation.”: “A wide variation was noted in the models, frameworks or processes of community organising utilised in these studies. We concluded that diversity implies that no single model, framework or process seems to have predominance over others in implementing community organising as a vehicle of positive social change within the health domain.”
Response to comment 3: The word “diversity” has now been replaced with “variation” in most places.
Introduction
Comment 4: This first sentence (“Community involvement in research…”) should be scrutinized further, since this is the hypothesis that the paper is testing. I would not state this with such certainty and remind readers that these are the supposed/alleged/assumed implications of community involvement. For example, are their shared goals of health improvement being met after these projects are concluded? The term “empower” is ambiguous here.
Response to comment 4: The sentence now reads- “Community involvement in research engages, empowers and mobilises people with an aim to achieve their shared goals by addressing structural inequalities in the social and built environment”
Results
Comment 5: Is it worth reiterating that only 38 papers were found? I am glad the authors found that many, but it seems like a low number by any standard, especially with a beginning pool of nearly five thousand?
Response to comment 5: 38 papers included in a review is considered an appropriate number given that the focus was on a health context.
Comment 6: Also, I wonder how the sample size would change if the community involvement outcomes were based on environmental (e.g., air/water/soil quality) interventions. I suspect the literature would have more examples given the propensity of citizen/community science initiatives, but I also wonder if the results of the evaluation would be similar (I suspect they might).
Response to comment 6: The results would no doubt change if we included other contexts such as environment, the researchers were specifically interested in the use of community organising for health interventions.
Comment 7: This, I think, should be sure to be reiterated as a main finding: “The reviewed studies did not document long-term outcomes or seem to document measurable changes in health outcomes.” Because it does beg the question of the initial premise of “achieving shared goals” – i.e., do they mean during the project, or the long-term afterward?
Response to comment 7: We have added the following: The reviewed studies did not document long-term outcomes or health impacts.
And:
Future studies should aim to measure long-term impact from their initiative, not just the measurement of outcomes during the funded period.
Comment 8: It’s okay and highly relevant to scrutinize the effectiveness of these efforts, since these efforts represent significant investments in time and resources from both communities and researchers, so looking for best practices requires a critical eye.
Response to comment 8: Thank you for this feedback. We have added to the discussion on the need for objective measures of initiative outcomes.
Table 1
Comment 9: Provide more information inn table caption; define each column header. I assume “S. No.” is “sample number” but what is the second number before author’s name? Also caveat/define “Intervention success,” and that this is based on the original study’s findings, and they were not judged by this review (also explain the ‘NA’ designation; did they not try to assess it, or were they inconclusive?)
Response to comment 9: More information has been added under the table. Column headers and abbreviations have now been defined. The numbers before author’s name was a formatting error, and has now been resolved.
Comment 10: Scrutiny of the original evaluations could be emphasized. For example, “building trust and generating commitment” (e.g., #4) is fairly subjective.
Response to comment 10: The analysis of the included studies’ outcomes indicates that there is room for improvement with the objectives and measured outcomes of the initiatives to be more specific with objective measures.
Comment 11: This is an incredible finding, again because of the time/resource commitment. One must question whether this is an appropriate timeframe. I realize this is leading up to the “action phase” but this is still a long time and seems accurate based on my experience: “Most studies took 24 months or more to start the action phase.”
Response to comment 11: Thank you for this feedback.
Comment 12: No Health Impact Assessments? (e.g., from PEW): https://www.pewtrusts.org/en/research-and-analysis/data-visualizations/2015/hia-map?sortBy=relevance&sortOrder=asc&page=1)
Response to comment 12: No health impact assessments were included in the studies.
Comment 13: So all the studies used steps, and about half of them followed an existing framework. (The review identified 22 different frameworks, models, or processes adopted. There were 10 studies that explicitly reported the use of a community organising framework. All studies reported their use of community organising steps, and these varied between four and 10 steps.)
Response to comment 13: Yes, all studies used steps.
Table 2
Comment 14: Somewhere, take a sentence or two to define “Framework/Model/Process” – this is not critical and does not need to be over-precise, but it would help to have a basic frame of reference.
Response to comment 14: Framework/model and process have been defined in the results section under the sub-heading 8.1 ‘Framework/model and process’.
8.1.3. CBPR
Comment 15: Mentioned above: **Wonder how results would have changed if ‘environmental’ was included in search parameters? Perhaps the sole focus on health is what yielded a seemingly low sample size (here and overall).
Response to comment 15: Please see above response 5.
Comment 16: Much about “power” and I believe this is typically defined, but could use as example for evaluation challenges (Kang [27] and Bauermeister[13] highlighted the shared power. Ross et al. [33] described the power of youth’s research and participation/action on health behaviour)
Response to comment 16: Thank you for this suggestion, we have included this in the discussion.
8.2 Targeted health behaviors
Comment 17: Again, on the topic of “short term” evaluations: “The outcomes of most initiatives were promising, with positive changes reported (at least in the short term) in health outcomes for the target populations in most studies (32/38).”
Response to comment 17: While most studies reviewed reported positive change in the health outcomes, we noted that these measures were typically collected over relatively short and focused project durations. Many studies did not discuss long term, sustainability of positive impacts, particularly after the funding had been exhausted.
Quality appraisal (this needs a heading number)
Number 7 has been included.
Discussion
Comment 18: Diversity – of options for community organizing
Change “Diversity” (mentioned above): A wide variety of models…. The diversity implies that no one specific model, framework or process seems to have predominance over others in implementing community organising
Response to comment 18: Diversity word has been replaced by variation.
Comment 19: I would debate whether the “consistent themes” are so different (“…some consistent themes were prevalent across the reviewed studies…”)? These themes are very similar and subjective. If nothing else, perhaps an acknowledgment of this observation, or at least that many of these are relative/subjective to whomever is defining it.
Interesting: “However, this review suggested that community organising as a vehicle of health initiatives or interventions has yet to pick up traction in countries outside of the United States.”
Response to comment 19: We have removed “consistent”.
Comment 20: Change “Diversity” (mentioned above): “Despite the heterogeneity in the selection of target population groups, there was consistency among most studies in terms of positive change reported in their targeted health outcomes. Such consistency in positive outcomes despite the diversity among target population groups reinforces the argument that community organising has the potential to be an important vehicle for positive change [12,13,20,25,27].”
Response to comment 20: Diversity word has been replaced by variation.
Comment 21: Perhaps a table that summarizes primary conclusions from Discussion and Limitations? (e.g., While most studies reviewed reported positive change in the health outcomes, we noted that these measures were typically collected over relatively short and focused project durations. Many studies did not discuss long term, sustainability of positive impacts, particularly after the funding had been exhausted.)
For example, if they had worked to change an existing program, policy, or decision (that affects health), would that have longer term impacts?
Response to comment 21: Thank you for this suggestion, the paper already is very long so we have chosen not to include any additional tables.
Reviewer 3 Report
The manuscript contains a high quality review with a detailed description of literature search mechanism, which makes the selection less arbitrary while more reproducible.
With a comprehensive selection of relevant works, the authors summarized eligible studies from various perspectives, including frameworks/models, study designs, target populations, health outcomes and a scoring tool to evaluate each study. Similarities, diversities and differences between studies are also discussed. Limitations such as lack of long term sustainable follow-up is also discussed. Current research activity disparities caused by health systems, political elements between countries are also addressed.
From an epidemiology perspective, there are some limitations to the study that are not discussed. Studies included in this paper in terms of endpoint/outcome selection can be categorized into four groups: 1). to identify issues (1/38), 2). description of program implementations (6/38), 3). evaluation of program implementations with lessons learnt and influencing factors (19/38), and 4). effect directly on health outcomes (12/38). Among the 12 studies directly on health outcomes, 5 discussed quantitative/statistical conclusions. So my suggestions are:
First, the four categories above mark the outcomes with increasing correlation to the ultimate objective: to improve health. Quality of implementation and effect on community capacity is an intermediate/surrogate outcome to the final health outcome. It would be nice if the lack of more direct health measurements are pointed out.
Second, A large variety of frameworks/models were identified and a wide spectrum of target populations were included. The rationale why each studies selected its specific framework/model to answer its population-specific questions can be an interesting topic. A consistent positive outcome of community organising process regardless of heterogenous background was noted. However, different community have unique starting points and challenges. Some may have poor baseline with huge room to benefit from community organising enhanced public health policies, while the room for others may be limited. This difference will determine the suitability of different initiatives, which can also be discussed.
Finally, the paper mentioned health promotion initiatives that engage communities are more likely to success. There is a lack of direct comparison to other initiatives without community engagement. The way how literature search was setup makes it difficult to include papers utilizing non-community-organising methods to address similar issues in similar target populations. A comparison to studies which do not engage communities can be an interesting topic.
In addition, some minor (editing) errors are also noticed. For instance, 1). Google scholar was mentioned but no records were identified from it. 2). theory, model, framework are the keyworks discussed, but the database search keywork omitted "model". 3). a huge percentage of records were excluded (4546 out of 4696) in the first step screening, but no rationale was given, which seems a little arbitrary. It can be a source of selection bias.
Again, it's a nice review which summarizes the current stage of research with extensive effort in material collections. A couple of minor discussions could make it more wholesome.
Author Response
The manuscript contains a high quality review with a detailed description of literature search mechanism, which makes the selection less arbitrary while more reproducible.
With a comprehensive selection of relevant works, the authors summarized eligible studies from various perspectives, including frameworks/models, study designs, target populations, health outcomes and a scoring tool to evaluate each study. Similarities, diversities and differences between studies are also discussed. Limitations such as lack of long term sustainable follow-up is also discussed. Current research activity disparities caused by health systems, political elements between countries are also addressed.
From an epidemiology perspective, there are some limitations to the study that are not discussed. Studies included in this paper in terms of endpoint/outcome selection can be categorized into four groups: 1). to identify issues (1/38), 2). description of program implementations (6/38), 3). evaluation of program implementations with lessons learnt and influencing factors (19/38), and 4). effect directly on health outcomes (12/38). Among the 12 studies directly on health outcomes, 5 discussed quantitative/statistical conclusions. So my suggestions are:
Comment 1: First, the four categories above mark the outcomes with increasing correlation to the ultimate objective: to improve health. Quality of implementation and effect on community capacity is an intermediate/surrogate outcome to the final health outcome. It would be nice if the lack of more direct health measurements are pointed out.
Response to comment 1: Thank you, we have added to the discussion to address this point:
Studies included in this review in terms of outcome selection can be categorized into four groups: 1). to identify issues (1/38), 2). description of program implementations (6/38), 3). evaluation of program implementations with lessons learnt and influencing factors (19/38), and 4). effect directly on health outcomes (12/38). However, among the 12 studies directly reporting on health outcomes, five discussed quantitative/statistical conclusions. These four categories show the outcomes with increasing correlation to the ultimate objective: to improve health. Quality of implementation and effect on community capacity is an intermediate outcome to the final health outcome. The inclusion of more direct health measurements would improve the ability to evaluate the impact of these initiatives. Future studies should aim to measure long-term impact from their initiative, not just the measurement of outcomes during the funded period.
Comment 2: Second, A large variety of frameworks/models were identified and a wide spectrum of target populations were included. The rationale why each studies selected its specific framework/model to answer its population-specific questions can be an interesting topic. A consistent positive outcome of community organising process regardless of heterogenous background was noted. However, different community have unique starting points and challenges. Some may have poor baseline with huge room to benefit from community organising enhanced public health policies, while the room for others may be limited. This difference will determine the suitability of different initiatives, which can also be discussed.
Response to comment 2: Thank you for this comment, we have added a future research suggestion to address this: While there a range of frameworks identified in this review, they are applied to different contexts, future research could examine the suitability of different frameworks for different community contexts taking into consideration their unique issues and starting points.
Comment 3: Finally, the paper mentioned health promotion initiatives that engage communities are more likely to success. There is a lack of direct comparison to other initiatives without community engagement. The way how literature search was setup makes it difficult to include papers utilizing non-community-organising methods to address similar issues in similar target populations. A comparison to studies which do not engage communities can be an interesting topic.
Response to comment 3: Thank you, we agree that comparing to studies that did not implement community organising approaches would be interesting, however for the purpose of this study we have decided not to do a comparison.
Comment 4: In addition, some minor (editing) errors are also noticed. For instance, 1). Google scholar was mentioned but no records were identified from it. 2). theory, model, framework are the keyworks discussed, but the database search keywork omitted "model". 3). a huge percentage of records were excluded (4546 out of 4696) in the first step screening, but no rationale was given, which seems a little arbitrary. It can be a source of selection bias.
Response to comment 4: Google scholar’s record were mentioned in Prisma flow chart. The keyword “model” was used during the literature search together with ‘framework/theory’ synonymously. The word “model” has now been added to the database search as well. The rationale for exclusion of studies has also been added.
Comment 5: Again, it's a nice review which summarizes the current stage of research with extensive effort in material collections. A couple of minor discussions could make it more wholesome.
Response to comment 5: Thank you, as suggested we have added to the discussion of the paper which has strengthened the contributions of the paper.